# Raman Characterization of the In-Plane Stress Tensor of Gallium Nitride

**DOI:** 10.3390/ma16062255

**Published:** 2023-03-10

**Authors:** Bowen Han, Mingyuan Sun, Ying Chang, Saisai He, Yuqi Zhao, Chuanyong Qu, Wei Qiu

**Affiliations:** 1Department of Mechanics, School of Mechanical Engineering, Tianjin University, Tianjin 300072, China; 2Tianjin Key Laboratory of Modern Engineering Mechanics, Tianjin 300072, China

**Keywords:** GaN, Raman spectroscopy, in-plane stress tensor, stress components, nanoindentation

## Abstract

Experimental characterization of the in-plane stress tensor is a basic requirement for the development of GaN strain engineering. In this work, a theoretical model of stress characterization for GaN using polarized micro-Raman spectroscopy was developed based on elasticity theory and lattice dynamics. Compared with other works, the presented model can give the quantitative relationship between all components of the in-plane stress tensor and the measured Raman shift. The model was verified by a calibration experiment under step-by-step uniaxial compression. By combining the stress characterization model with the expanding cavity model, the in-plane residual stress component field around Berkovich indentation on the (0001) plane GaN was achieved. The experimental results show that the distributions of the stress components, which significantly differed from the distribution of the Raman shift, were closely related to the GaN crystal structure and exhibited a gradient along each crystal direction.

## 1. Introduction

Gallium nitride (GaN) is regarded as one of the representatives of the third generation of semiconductor materials due to its fascinating intrinsic properties, such as a wide direct band gap (3.4 eV), strong binding energies and excellent thermal stability and conductivity [1,2], which are widely employed in light-emitting diodes (LEDs), high-electron-mobility transistors and high-temperature, high-frequency and high-power semiconductor devices [3,4,5]. Semiconductor devices using GaN materials and related strain engineering require the introduction of state- and size-controllable stress in the GaN layer in the manufacturing process to regulate device functions and to improve the quality and reliability of the devices. Therefore, it is necessary to accurately measure and analyze the complex stress state of GaN materials.

Micro-Raman spectroscopy (MRS) is an effective method to achieve high-resolution [6,7] and nondestructive [8,9,10] detection of stress in materials. Early studies of light-scattering behavior in condensed matter physics found that the Raman spectra of a variety of crystal materials were affected by stress/strain. De Wolf et al. [11,12] gave a simplified quantitative relationship between the Raman shift and stress in single-crystal silicon (Si) and characterized the stress introduced by different processes in a variety of microelectronic structures based on MRS. Lei and Kang et al. [13,14] applied MRS to systematically analyze the characterization model and distribution law of residual stress in porous silicon. Xu et al. [15] investigated the interfacial mechanical parameters of graphene by MRS, and discovered for the first time that there were significant spatial/time scale effects on these interfacial mechanical parameters. Zhao et al. [16,17] carried out experimental studies on the crack extension and fracture toughness of graphene and the interfacial mechanical behavior of bilayer graphene using MRS.

In recent years, with the development of GaN semiconductor devices, a series of advances have been made in research on the Raman stress characterization of GaN materials. For example, Briggs et al. [18] gave a relationship between the Raman shift and strain of each mode of GaN and a solution formula for the deformation potential constants of each mode based on linear deformation potential theory. Davydov et al. [19] obtained specific values of deformation potential constants of the partial mode for GaN based on the bulk Grueneisen parameter of the Raman shift of mode during hydrostatic compression under the condition of disregarding the effect of shear strain. Darakchieva et al. [20] explored the relationship between the anisotropic strain of GaN and the mode deformation potential by combining the elliptical polarization method, and calculated the specific value of the deformation potential constants of the mode associated with the tangential strain. Schustek et al. [21] constructed a linear set of equations for the Raman shift–strain relationship for different modes by linearly simplifying the Raman shift–strain equation. Amilusik et al. [22] gave a simplified model of the stress characterization for GaN using the property of the doubly degenerate E_1_ and E_2_ modes under anisotropic strain.

Based on the above basic research results, Amilusik et al. [22] characterized the residual stress field of ammonothermal grown GaN on substrates in lateral and vertical direction. Hossain et al. [23] analyzed the residual stress in GaN films on Si substrates of different sizes and determined that the stress state of GaN films could be modulated by varying the space size of the Si substrate. Wang et al. [24] investigated the effect of high temperature and mechanical stress on the breakdown voltage of GaN devices and found that both temperature and mechanical stress reduced the performance of the devices by decreasing the breakdown voltage. Park et al. [25] characterized the stress state of GaN films on different substrates, revealing the potential of lithium silicate as an ideal substrate. Chai et al. [26] analyzed the gradient of in-plane residual stress existing in gallium nitride/silicon carbide (GaN/SiC) microstructures of different sizes and found that the residual stress in the GaN film is proportional to the residual stress in the SiC buffer layer and that the stress state in the GaN film can be regulated by modulating the stress state in the SiC layer.

In the published work, the characterization model for Raman stress analysis of GaN could only give the analytical relationship between the sum of the principal stresses and the Raman shift. Moreover, the previous model did not consider the effect of each stress component and could not provide detailed information about the stress state. To address the above problems, this paper developed a model for the stress characterization of GaN based on polarized Raman spectroscopy and established an analytical relationship between the stress component and the measured polarized Raman shift. Our derived model was verified by a calibration experiment to realize the accurate characterization of the complex stress field on the GaN surface.

## 2. Materials and Experiments

The sample used in this paper was an undoped N-GaN (negative-gallium nitride) single crystal (Galactic Semiconductor Technology Co., Ltd., Anhui, China). The sample parameters are listed as follows: thickness of 0.374 mm, crystal plane pointing to the (0001) plane, main reference plane of the ($1\overset{-}{1}00$) plane, dislocation density < 2 × 10^6^ cm^−2^ and etch pit density < 1.5 cm^−2^. The 2-inch GaN sample was cut into several rectangular-shaped samples of 10 mm × 2 mm and 10 mm × 10 mm in length and width, respectively, using a scribing machine.

In this paper, uniaxial compression experiments were performed on GaN samples (10 mm × 2 mm × 0.374 mm) using a miniature in situ mechanics tester (CARE IBTC-300S, Care Measurement & control Co., Ltd., Tianjin, China), as shown in Figure 1a. The displacement resolution and load resolution of the mechanics tester were approximately 0.1 μm and 0.1 mN, respectively. The compression experiment was performed with a step loading of 5 N and loaded to 70 N. The Raman scattering signal from the sample surface was acquired three times after each loading using a laser confocal micro-Raman spectrometer (WITEC Alpha 300R, WITec Co., Ulm, Germany), as shown in Figure 1b. The spectral resolution and spectral stability of the Raman spectrometer were approximately 0.8 cm^−1^ and ±0.02 cm^−1^, respectively. The spatial resolution of the displacement stage attached to the Raman spectrometer was approximately 50 nm. When acquiring the Raman scattering signal, the spectrometer parameters were set as follows: 50× objective (numerical aperture, N.A. = 0.55), laser wavelength of 532 nm, laser power of 20 mW, exposure time of 3 s, grating of 1800 L/mm and parallel polarization configuration (polarization directions of the incident laser parallel to the polarization direction of the scattered light).

Nanoindentation experiments were performed on a GaN sample (10 mm × 10 mm × 0.374 mm) using a nanoindentation instrument (Agilent Nano Indenter G200, Agilent Technology Inc., Santa Clara, CA, USA), as shown in Figure 1c. The displacement resolution and load resolution of the nanoindentation instrument were approximately 0.01 nm and 50 nN, respectively. A Berkovich indenter with a radius of curvature of 50 nm was used to squeeze the GaN (0001) surface vertically at room temperature, with one side of the indenter parallel to the ($1\overset{-}{1}00$) plane of the sample. The nanoindentation experimental parameters are listed as follows: loading rate of 10 nm/s, allowable drift rate of 0.55 nm/s and maintenance of the maximum load state for 10 s before unloading. The final indentation depth of the nanoindentation experiment was 1300 nm. Then, a Raman mapping experiment was performed around the indentation. The Raman mapping experimental parameters are listed as follows: 100 × objective (N.A. = 0.90), laser wavelength 532 nm, output power of the laser device 20.0 mW (the actual laser power reaching the sample surface in the experiment was about 6.3 mW), exposure time 1 s, grating 1800 L/mm and parallel polarization configuration. The mapping area was a 50 μm × 50 μm square area centered on the indentation, with horizontal and vertical scan steps of 0.5 μm. The schematic is shown in Figure 2.

## 3. Analysis and Discussion

### 3.1. Model

GaN crystals have a wurtzite structure (as shown in Figure 3) and belong to the hexagonal system. Its lattice group is C_6v_^4^, and the basic unit cell is composed of four atoms. According to the factorial group analysis, the following relationship exists for the optical phonon modes of wurtzite GaN at the Γ point in the Brillouin zone [27]:(1)$$\Gamma_{opt}=\Gamma_{1}+\Gamma_{4}+\Gamma_{5}+\Gamma_{5}=A_{1}\left(z\right)+2B+E_{1}\left(x,y\right)+2E_{2}$$
where the A_1_, E_1_ and E_2_ (E_2_^low^ and E_2_^high^) modes are Raman active and the B mode is silent. Due to the macroscopic electric field associated with the relative atomic displacement of the longitudinal modes, the A_1_ and E_1_ modes are split into longitudinal optical (LO) components and transverse optical (TO) components [28].

Amilusk’s study showed that the E_1_(TO) and E_2_^high^ modes are sensitive to biaxial stress in the plane perpendicular to the *c*-axis, the A_1_(TO) mode is sensitive to uniaxial stress in the direction parallel to the *c*-axis and the E_2_^low^ mode is rather insensitive to biaxial stress and hydrostatic compression [22]. Figure 4 shows a typical Raman spectrum of the (0001) plane GaN we measured in the vertical backscattering configuration. The characteristic peak corresponding to the A_1_(TO), A_1_(LO), E_2_^low^ and E_2_^high^ modes is observed. The characteristic peak corresponding to the E_1_ mode is not observed, as the E_1_ mode is forbidden due to the Raman selection rule in the vertical backscattering configuration of the (0001) plane. Since the signal intensity of the E_2_^high^ mode is higher than that of the E_2_^low^ mode, we chose the E_2_^high^ mode to establish an analytical relationship between the Raman shift and the stress component.

The general process of establishing the relationship between the Raman shift and stress is shown in Figure 5 [29].

Expanding the lattice dynamics equation along the Cartesian coordinate system and setting the determinant of its coefficients equal to zero, the characteristic equation of lattice dynamics is obtained:(2)$$\left|\begin{array}{ccc} \varepsilon_{uv}K_{uv11}-\lambda & \varepsilon_{uv}K_{uv12} & \varepsilon_{uv}K_{uv31}\\ \varepsilon_{uv}K_{uv21} & \varepsilon_{uv}K_{uv22}-\lambda & \varepsilon_{uv}K_{uv32}\\ \varepsilon_{uv}K_{uv31} & \varepsilon_{uv}K_{uv32} & \varepsilon_{uv}K_{uv33}-\lambda \end{array}\right|=0,\;u,v=1,\;2,\;3$$
where *ε_uv_* is the component of the strain tensor ***ε*** and *K_uvij_* (*i*, *j* = 1, 2, 3) is a component of the mode variable state tensor ***K***, which is used to describe the change in the elastic constant caused by strain. According to the generalized Hooke’s law, there is a linear relationship between stress and strain in the case of small deformation:(3)ε=Sσ
where ***σ*** is the stress tensor and ***S*** is the elasticity tensor. Introducing Equation (3) to Equation (2), the eigenvalues *λ_k_* (*k* = 1, 2, 3) of the characteristic equation of lattice dynamics can be solved. For the eigenvalues *λ_k_*, there exists the relationship $\lambda_{k}={\;\omega }_{k}^{2}-{\;\omega }_{0}^{2}$, where $\omega_{0}$ and $\omega_{k}$ are the Raman shifts of the Raman vibration mode in the stress-free state and stressed state, respectively. In the case of small deformation, the Raman shift increment $\Delta \omega_{k}$ is much smaller than $\omega_{0}$, so the following relationship exists:(4)$$\Delta \omega_{k}=\omega_{k}-\omega_{0}\approx \frac{\omega_{k}^{2}-\omega_{0}^{2}}{2\omega_{0}}=\frac{\lambda_{k}}{2\omega_{0}}$$

In the actual measurement process, the intensity of each characteristic peak depends on the Raman selection rule, that is, the Raman intensity *I* of the Raman vibrational mode is related to the Raman tensor and the polarization vector of the incident laser and scattered light, as shown in Equation (5):(5)$$I=Q\sum_{k}I_{k}=Q{\sum_{k}|e_{I}^{T}\cdot R_{k}\cdot \;e_{S}|}^{2},\;\;\;k=1,\;2,\;3$$
where *Q* is a constant determined by the scattering law of the sample, the optical properties and the experimental equipment; $e_{I}$ and $e_{S}$ are the polarization vectors of the incident laser and scattered light, respectively; and $R_{k}$ is the Raman tensor of the *k* mode. The actual measured Raman shift $\Delta \omega_{\mathrm{obs}}$ after the deformation is a linear combination of the mode Raman shift $\Delta \omega_{k}$ within the same characteristic peak weighted by its respective contributions to the total scattered intensity:(6)$$\Delta \omega_{obs}=\frac{\sum_{k=1}^{3}\Delta \omega_{k}I_{k}}{\sum_{k=1}^{3}I_{k}}$$

For GaN, the sample coordinate system (overlapping with the crystal coordinate system) was established with the [$11\overset{-}{2}0$] direction as the *x*-axis, the [$\overset{-}{1}100$] direction as the *y*-axis and the [0001] direction as the *z*-axis. Since the E_2_ mode of GaN is doubly degenerate [22], the characteristic equation of lattice dynamics is expressed as:(7)$$\left|\begin{array}{cc} a_{E_{2}}(\varepsilon_{x}+\varepsilon_{y})+b_{E_{2}}\varepsilon_{z}+c_{E_{2}}(\varepsilon_{x}-\varepsilon_{y})-\lambda & 2c_{E_{2}}\varepsilon_{xy}\\ 2c_{E_{2}}\varepsilon_{xy} & a_{E_{2}}(\varepsilon_{x}+\varepsilon_{y})+b_{E_{2}}\varepsilon_{z}-c_{E_{2}}(\varepsilon_{x}-\varepsilon_{y})-\lambda \end{array}\right|=0$$
where $a_{E_{2}}$, $b_{E_{2}}$ and $c_{E_{2}}$ are the phonon deformation potential constants of the E_2_ mode [19,20]. Combining Equations (4) and (7), the following relationship is obtained:(8)$$\left\{ \begin{array}{l} \;\Delta \omega_{1}=\left[a_{E_{2}}(\varepsilon_{x}+\varepsilon_{y})+b_{E_{2}}\varepsilon_{z}+c_{E_{2}}\sqrt{{\left(\varepsilon_{x}-\varepsilon_{y}\right)}^{2}+4\varepsilon_{xy}^{2}}\right]/2\omega_{0}\\ \;\Delta \omega_{2}=\left[a_{E_{2}}(\varepsilon_{x}+\varepsilon_{y})+b_{E_{2}}\varepsilon_{z}-c_{E_{2}}\sqrt{{\left(\varepsilon_{x}-\varepsilon_{y}\right)}^{2}+4\varepsilon_{xy}^{2}}\right]/2\omega_{0} \end{array}\right.$$

The Raman tensor corresponding to the E_2_ mode is [30]:(9)$$R_{1}=\left(\begin{array}{ccc} 0 & d & 0\\ d & 0 & 0\\ 0 & 0 & 0 \end{array}\right),\;\;\;R_{2}=\left(\begin{array}{ccc} d & 0 & 0\\ 0 & -d & 0\\ 0 & 0 & 0 \end{array}\right)$$
where *d* is a constant that depends on the Raman polarization of the sample.

The measured GaN surface is in the state of in-plane stress:(10)$$\sigma ={\left[\sigma_{x}\sigma_{y}\;0\;\tau \;0\;0\right]}^{T}$$
where *σ_x_* and *σ_y_* are the normal stress components of the stress tensor ***σ*** in the sample coordinate system along the *x* direction and *y* direction, respectively, and *τ* is the shear stress component in the *x*-*y* plane. Combining Equations (3) and (10), the stress–strain equation for GaN in the in-plane stress state is obtained:(11)$$\left[\begin{array}{c} \varepsilon_{x}\\ \varepsilon_{y}\\ \varepsilon_{z}\\ \varepsilon_{xy}\\ \varepsilon_{yz}\\ \varepsilon_{zx} \end{array}\right]=S\sigma =\left[\begin{array}{cccccc} S_{11} & S_{12} & S_{13} & & & \\ S_{12} & S_{11} & S_{13} & & & \\ S_{13} & S_{13} & S_{33} & & & \\ & & & S_{44} & & \\ & & & & S_{44} & \\ & & & & & S_{66} \end{array}\right]\left[\begin{array}{c} \sigma_{x}\\ \sigma_{y}\\ 0\\ \tau \\ 0\\ 0 \end{array}\right]=\left[\begin{array}{c} S_{11}\sigma_{x}+S_{12}\sigma_{y}\\ S_{12}\sigma_{x}+S_{11}\sigma_{y}\\ S_{13}\left(\sigma_{x}+\sigma_{y}\right)\\ S_{44}\tau \\ 0\\ 0 \end{array}\right]$$

By combining Equations (3), (8), (9) and (11), the relationships between the stress component and the Raman shift of (0001) plane GaN in the polarization configuration is obtained:(12)$$\left\{ \begin{array}{l} \;\mathrm{HH}\;\Delta \omega_{obs}=-1.233\left(\sigma_{x}+\sigma_{y}\right)-\left(1.554\cos4\alpha \right)\sqrt{{\left(\sigma_{x}-\sigma_{y}\right)}^{2}+21.475\tau^{2}}\\ \;\mathrm{HV}\;\Delta \omega_{obs}=-1.233\left(\sigma_{x}+\sigma_{y}\right)+\left(1.554\cos4\alpha \right)\sqrt{{\left(\sigma_{x}-\sigma_{y}\right)}^{2}+21.475\tau^{2}} \end{array}\right.$$
where HH and HV are the polarization directions of the incident laser, parallel and perpendicular, respectively, to the polarization direction of the scattered light, and *α* is the angle of polarization of the incident laser.

### 3.2. Calibration Experiment

In this paper, uniaxial compression experiments were conducted using GaN samples to verify the above model. In the stress-free state, the Raman wavenumber of the E_2_^high^ mode we measured is 568.2 cm^−1^, which is close to the value measured by others [31]. The Raman shift of the E_2_^high^ mode is the difference between the Raman wavenumber when the E_2_^high^ mode is stressed and that when it is stress-free. For the experimental data analysis, we used the three Raman spectra acquired at each load value to calculate the Raman shift separately and averaged them as average Raman shift values. The stress values obtained from the step-loading measurement and the average Raman shift values of the E_2_^high^ mode at this stress value were plotted as the horizontal and vertical coordinates of the experimental data graph, and then the slope (Raman shift–stress coefficient) was obtained by linear fitting to the experimental measurement points, as shown in Figure 6. From the fitted results, the Raman shift–stress coefficient of the (0001) plane GaN in the uniaxial stress state is −2.69 cm^−1^/GPa. Based on the relationship between the in-plane stress and the Raman shift of (0001) plane GaN in the HH configuration in Equation (12), the Raman shift–stress coefficient of the E_2_^high^ mode in the same stress state is −2.79 cm^−1^/GPa. The comparison shows that the relative error between the experimental results and the theoretical results is less than 3.6%, so the experimental results better verify the correctness of our derived theoretical model.

Other scholars had also measured the value of the Raman shift–stress coefficient by experiments. Demangeot et al. [32] gave a value of −2.9 cm^−1^/GPa for the Raman shift–stress coefficient, which is closer to our experimental result. This is because the stress state in this work is consistent with the stress state of this paper (uniaxial compressive stress state). The value of the Raman shift–stress coefficient measured by Kisielowski et al. [33] is −4.2 cm^−1^/GPa, which differs from our experimental result. The reason is that the stress state of the measured sample, as well as some material constants used in Raman stress model, are different from those in this paper. Kozawa et al. [34] obtained a value of −6.2 cm^−1^/Gpa for the Raman shift–stress coefficient, which is much larger than that obtained in this work. The main reason is that the Olsen and Ettenberg model used in Kozawa’s work establishes the relationship between the film stress and the bending deformation based on the numbers of material and structural idealizations. It is not applicable to multilayer problems with unknown or variable elastic parameters or thickness or stress state. The sample in this work had a buffer layer neglected by the authors. Kisielowski et al. [33] showed that the buffer layer affects the residual stress state in the GaN layer.

### 3.3. Raman Shift–Stress Relationship around the Indentation

The study by Yoffe et al. showed that the stress around the indentation is consistent with the expanding cavity model constructed based on the cylindrical coordinate system [35,36,37,38]. Figure 7a gives a microphotograph of a nanoindentation on the (0001) plane GaN sample, and Figure 7b shows the sample coordinate system, crystal coordinate system and cylindrical coordinate system established at the surface center of the nanoindentation sample. In the cylindrical coordinate system, the *z* direction coincides with the axial direction of the cylindrical coordinate system in the (0001) plane, *r* denotes the radial direction and its anticlockwise angle with the *x*-axis is the tangential direction *φ*.

Based on the expanding cavity model, each component of the in-plane stress state at any point around the indentation is expressed as follows [37]:(13)$$\begin{array}{l} \sigma_{rr}=\frac{4B_{rr}}{r^{3}}\left(\nu_{r\varphi }-2\right),\;\sigma_{\varphi \varphi }=\frac{4B_{\varphi \varphi }}{r^{3}}\left(1-2\nu_{\varphi r}\right),\;\sigma_{r\varphi }=0\\ B_{rr}=0.102\frac{G_{rz}f_{\varphi \varphi }\sqrt[{3}]{P/H}}{\pi \left(1-2\nu_{\varphi r}\right)},\;B_{\varphi \varphi }=0.102\frac{G_{z\varphi }f_{\varphi \varphi }\sqrt[{3}]{P/H}}{\pi \left(1-2\nu_{\varphi r}\right)} \end{array}$$
where *r*, *φ* and *z* denote the radial direction, tangential direction and axial direction, respectively, under the cylindrical coordinate system; *σ_rr_* is the radial stress component, *σ_φφ_* is the tangential stress component and *σ_rφ_* is the shear stress component; *ν_rφ_* and *ν_φr_* are Poisson’s ratios and *G_rz_* and *G_zφ_* are the shear moduli, which can be calculated using the elasticity tensor $S^{\prime }$ in the cylindrical coordinate system; *B_rr_* and *B_φφ_* are functions characterizing the degree of material bulging caused by plastic deformation near the indentation; *f_φφ_* is a function characterizing the nonconservative change in volume after plastic deformation of the material near the indentation; *P* is the peak load applied to the indenter; and *H* is the Vickers hardness of the crystal. Since the radial shear stress component is 0, the stress at any point around the indentation given by the expanding cavity model is actually in a state of non-equal biaxial stress, and the directions of each principal stress are known. The ratio of the radial stress component to the tangential stress component is:(14)$$\frac{\sigma_{rr}}{\sigma_{\varphi \varphi }}=\frac{Grz\left(\nu r\varphi -2\right)}{Gz\varphi \left(1-2\nu \varphi r\right)}$$

From the definition of Poisson’s ratio and the shear modulus, it follows that:(15)$$\nu_{r\varphi }=-\frac{{S^{\prime }}_{12}}{{S^{\prime }}_{11}};\nu_{\varphi r}=-\frac{{S^{\prime }}_{21}}{{S^{\prime }}_{22}};G_{rz}=\frac{1}{{S^{\prime }}_{66}};G_{z\varphi }=\frac{1}{{S^{\prime }}_{55}}$$
where $S_{11}^{\prime }$, $S_{12}^{\prime }$, $S_{21}^{\prime }$, $S_{22}^{\prime }$, $S_{55}^{\prime }$ and $S_{66}^{\prime }$ are the components of the elasticity tensor $S^{\prime }$ in the cylindrical coordinate system.

Combining Equations (14) and (15), and substituting the material parameters of the crystal plane, the ratio of the radial to the tangential stress component is obtained as follows:(16)$$\frac{\sigma_{rr}}{\sigma_{\varphi \varphi }}|{}_{\left(0001\right)}=-\frac{\left(13.616+\cos2\varphi \right)\left(33-\cos4\varphi \right)}{\left(13.616-\cos2\varphi \right)\left(5.77+\cos4\varphi \right)}$$

Combined with the rotation axis equation of the stress component and Equation (12) in this paper, the relationship between the measured Raman shift of the E_2_ mode of GaN in the HH configuration and *σ_rr_* and *σ_φφ_* in the cylindrical coordinate system is obtained:(17)$$\Delta \omega_{\mathrm{obs}}=-1.233\left(\sigma_{\varphi \varphi }+\sigma_{rr}\right)-\left(2.297\cos4\alpha \right)\sqrt{\left(1.458-\cos4\varphi \right){\left(\sigma_{\varphi \varphi }-\sigma_{rr}\right)}^{2}}$$

Combining Equation (16) with Equation (17), the relationship between each stress component and the Raman shift at any position in the region around the GaN indentation is obtained as follows:(18)$$\begin{array}{l} \sigma_{\varphi \varphi }={\left\{\begin{array}{l} -\frac{\left(57.143+5.976\cos2\varphi -4.198\cos4\varphi \right)}{\left(13.616-\cos2\varphi \right)\left(5.77+\cos4\varphi \right)}+\left(\cos4\alpha \right)\\ \frac{\left(608.556+31.287\cos2\varphi -2.311\cos2\varphi \cos4\varphi \right)}{\left(13.616-\cos2\varphi \right)\left(5.77+\cos4\varphi \right)}\sqrt{1.458-\cos4\varphi } \end{array}\right\}}^{-1}\cdot \Delta \omega_{\mathrm{obs}}\\ \sigma_{rr}={\left\{\begin{array}{l} \frac{\left(57.143+5.976\cos2\varphi -4.198\cos4\varphi \right)}{\left(13.616+\cos2\varphi \right)\left(33-\cos4\varphi \right)}-\left(\cos4\alpha \right)\\ \frac{\left(608.556+31.287\cos2\varphi -2.311\cos2\varphi \cos4\varphi \right)}{\left(13.616+\cos2\varphi \right)\left(33-\cos4\varphi \right)}\sqrt{1.458-\cos4\varphi } \end{array}\right\}}^{-1}\cdot \Delta \omega_{\mathrm{obs}} \end{array}$$

### 3.4. Distribution of Residual Stress Component around the Indentation

In this paper, a Raman mapping experiment was performed on the region around the indentation of the GaN sample. The Raman peak of the E_2_^high^ mode was fitted using the Lorenz function, and the experimental data were analyzed by calculating the Raman shift of the E_2_^high^ mode. The distribution of the Raman shift of the E_2_^high^ mode around the (0001) plane indentation is shown in Figure 8. The distribution diagram of the Raman shift shows that there is a region of increasing shift near the indentation with a certain directionality. Specifically, the Raman shift significantly increases at the center of the indentation along the [$2\overset{-}{1}\overset{-}{1}0$], [$11\overset{-}{2}0$] and [$\overset{-}{1}2\overset{-}{1}0$] directions and decreases along the direction away from the center of the indentation with a gradient profile similar to that of a hexagonal system profile. The distribution of the values of the Raman shift shows that the values are all positive, indicating that the residual stress field is dominated by compressive stress. The extreme value of the Raman shift appears on the right side of the vertical side of the indentation near the center, which is approximately 4.0 cm^−1^. The Raman shifts along the [$2\overset{-}{1}\overset{-}{1}0$], [$11\overset{-}{2}0$] and [$\overset{-}{1}2\overset{-}{1}0$] directions show obvious gradients, where the Raman shifts along the [$2\overset{-}{1}\overset{-}{1}0$] and [$\overset{-}{1}2\overset{-}{1}0$] directions slowly decay and those along the [$11\overset{-}{2}0$] direction decay at a faster rate. The Raman shift basically decays to 0 cm^−1^ in most regions far from the indentation center at 20 μm, that is, the sample tends to be stress-free in these regions. From the overall distribution of the Raman shift, the distribution is symmetric along the [$2\overset{-}{1}\overset{-}{1}0$], [$11\overset{-}{2}0$] and [$\overset{-}{1}2\overset{-}{1}0$] directions.

By substituting the measured Raman shift into Equation (18), the distribution of the residual stress components in the region around the indentation of the (0001) plane GaN are obtained, as shown in Figure 9. The residual stress components have obvious distribution characteristics along the [$2\overset{-}{1}\overset{-}{1}0$], [$\overset{-}{1}2\overset{-}{1}0$], [$1\overset{-}{1}00$] and [$11\overset{-}{2}0$] directions, indicating that the distribution trend is related to the lattice structure of the crystalline material. Relatively, the gradient of stress variation along the [$1\overset{-}{1}00$] and [$11\overset{-}{2}0$] directions is more obvious. The distribution of stress values indicates that *σ_rr_* is negative (indicating compressive stresses) and that *σ_φφ_* is positive (indicating tensile stresses). The extreme values of *σ_rr_* and *σ_φφ_* appear on the right side of the vertical side of the indentation near the center, which are approximately −4.20 GPa and 0.90 GPa, respectively. At the same point of the sample, the value of *σ_rr_* is several times larger than that of *σ_φφ_*, further confirming the dominance of residual compressive stresses on the sample surface.

According to the distributions of *σ_rr_* and *σ_φφ_* in Figure 9, the distributions of normal stresses *σ_x_* and *σ_y_* and shear stress *τ* on the (0001) plane GaN in the rectangular coordinate system can be obtained by using the rotation axis equation, as shown in Figure 10. It should be noted that *σ_z_* was not considered when measuring the free surface of GaN. The distribution trend of the normal stresses in the region around the indentation is related to the lattice structure of the crystalline material. Both of them have positive and negative values. Specifically, the values of *σ_x_* along the [$2\overset{-}{1}\overset{-}{1}0$], [$\overset{-}{1}2\overset{-}{1}0$] and [$1\overset{-}{1}00$] directions are positive, and the values along the [$11\overset{-}{2}0$] direction are negative. The values of *σ_y_* along the [$2\overset{-}{1}\overset{-}{1}0$], [$\overset{-}{1}2\overset{-}{1}0$] and [$11\overset{-}{2}0$] directions are positive, and the values along the [$1\overset{-}{1}00$] direction are negative. At the proximal end of the indentation, *σ_x_* and *σ_y_* are mainly distributed along the [$1\overset{-}{1}00$] and [$11\overset{-}{2}0$] directions, and both have less variation in the values along the [$11\overset{-}{2}0$] direction. At the distal end of the indentation, *σ_x_* and *σ_y_* are mainly distributed along the [$2\overset{-}{1}\overset{-}{1}0$], [$11\overset{-}{2}0$] and [$\overset{-}{1}2\overset{-}{1}0$] directions. The values of *σ_rr_* and *σ_φφ_* show a distinct gradient along the [$2\overset{-}{1}\overset{-}{1}0$] and [$\overset{-}{1}2\overset{-}{1}0$] directions. The values of the two stress components converge to 0 GPa in most areas 20 μm from the indentation center. According to the distribution characteristics of the stress value, the two stress components are symmetrical about the [$1\overset{-}{1}00$] and [$11\overset{-}{2}0$] directions. For the shear stress, it has positive and negative values.

As we previously mentioned, most scholars have applied the relationship between the Raman shift and the sum of the principal stresses for stress characterization of GaN. A comparison of Figure 8 with Figure 9 shows that there is a large difference between the distribution of the Raman shift and the distribution of the stress components, which illustrates the limitations of the previous analytical ideas of stress characterization. If we also use the previous method, then we will obtain a stress distribution consistent with the distribution of the Raman shift in Figure 8. This approach will not only give the incorrect trend of stress distribution but also increase the numerical level of the overall stress value with the calculation error up to 174%. In addition, that method cannot give the directions of true stress at any point of the sample. Using our derived model for stress characterization in combination with the expanding cavity model, it is possible to give the stress values and stress directions at any point in the region around the indentation on the (0001) plane GaN. Therefore, the advantage of the stress characterization model around the indentation obtained by considering the stress state on the sample surface is that it can give the true stress information and then realize the fine characterization of the stress component field.

The above discussion proved that it should be meaningful to consider the influence of stress state when establishing the Raman stress model of GaN. If the stress state of the sample was not considered, that is, using the relationship between the Raman shift and the sum of the principal stresses to analyze the stress distribution trend, it would not only give an incorrect conclusion, but also achieve an error of stress value up to 174%. Meanwhile, based on the model presented in this work, the decoupling of the stress components of the (0001) plane GaN can be realized under the condition that the principal stress direction is known, which will be beneficial for further studies to realize the complete decoupling of the stress components of GaN in any arbitrary stress state. At the same time, the GaN stress characterization model proposed in this paper can solve most of the engineering problems. This is because in practical engineering problems, there usually exist positions with known stress states (such as free boundaries) which can be used as boundary conditions. Then the continuity relationship given by elastic mechanics should be used to derive the principal stress value and its direction at the analyzed location, thus realizing stress decoupling. Of course, such an analysis is required to combine Raman measurement with the mathematical or simulation modeling of elastic mechanics, hence it is not a direct measurement. Moreover, this paper can provide an analytical idea for the establishment of a Raman stress model of other material. In addition, the GaN samples we used in our experiments were not intentionally doped, so we did not consider the effect of doping in the analysis of stress, nor in the theoretical model.

## 4. Conclusions

This work, by establishing the GaN stress characterization model under polarized Raman, presented the relationship between the Raman shift and each stress component. Compared with the traditional method using the relationship between the Raman shift and the sum of the principal stresses, the proposed model considered the influence of stress state. The Raman shift–stress coefficient was achieved by the calibration experiment. By combining the stress characterization model with the expanding cavity model, the residual stress component field around the indentation was achieved based on Raman experiments in the HH configuration. The distributions of the residual stress components showed a close correlation with the crystal structure of the sample material, furtherly verified the importance of considering the stress state. This work would be beneficial to the strain engineering and stress characterization of GaN materials and their related devices.

## Figures and Tables

**Figure 1 materials-16-02255-f001:**
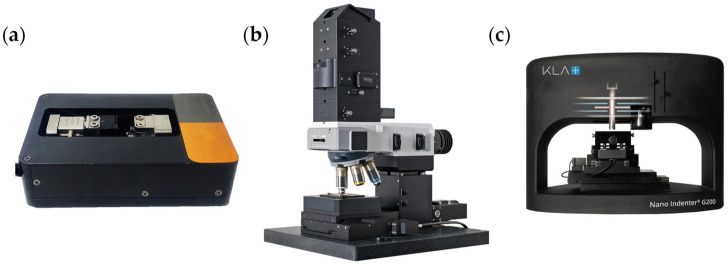
(**a**) Miniature in situ mechanics tester; (**b**) laser confocal micro-Raman spectrometer; (**c**) nanoindentation instrument.

**Figure 2 materials-16-02255-f002:**
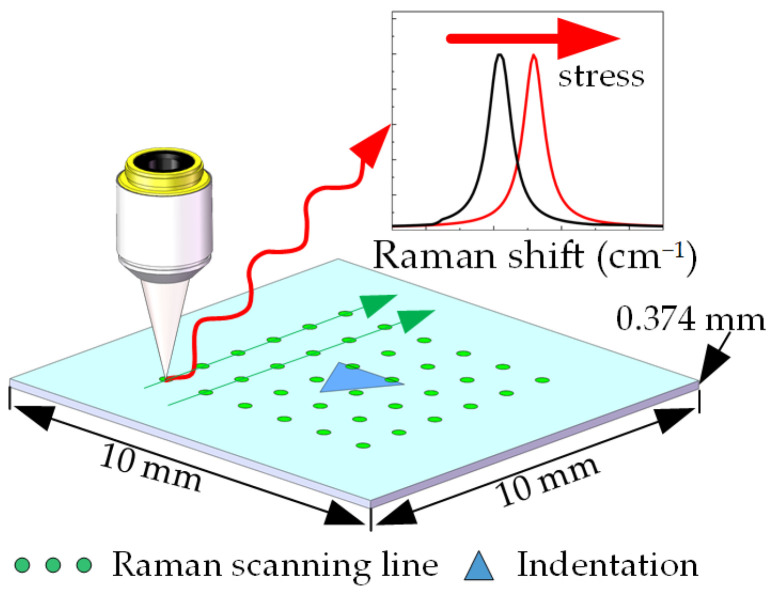
Schematic of the Raman mapping experiment around the indentation on the (0001) plane GaN.

**Figure 3 materials-16-02255-f003:**
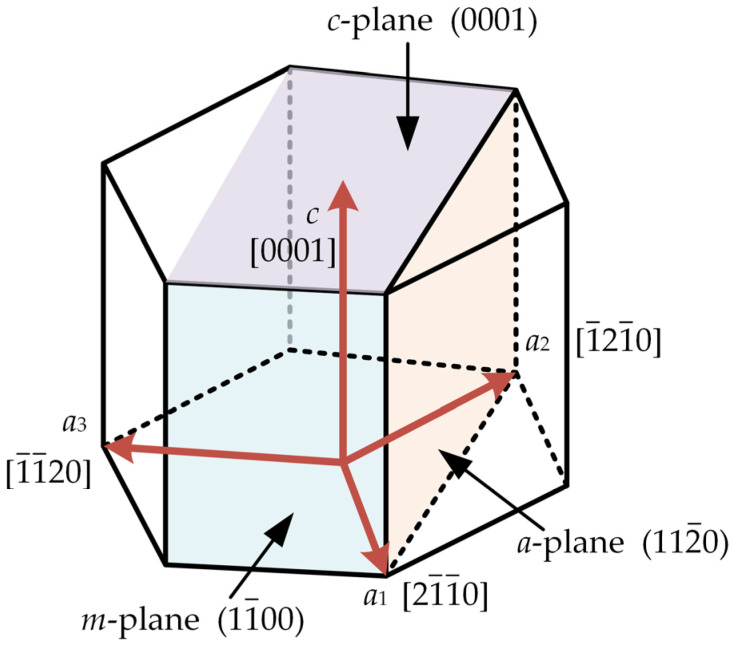
Schematic of the structure of wurtzite, where *a*_1_, *a*_2_, *a*_3_ and *c* denote the crystal orientations.

**Figure 4 materials-16-02255-f004:**
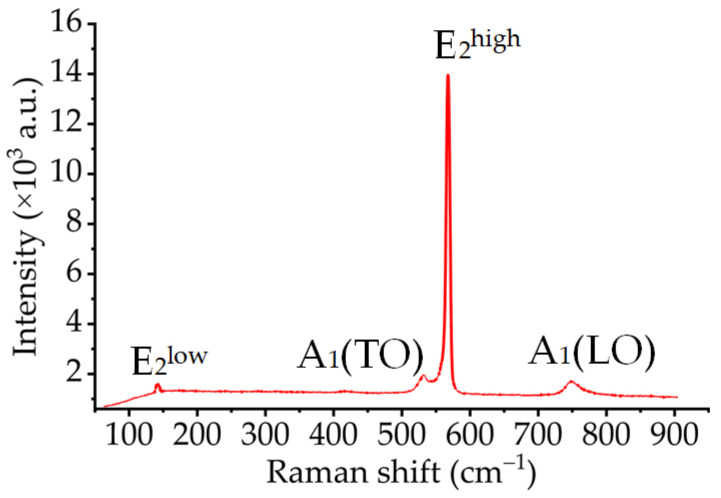
Raman spectrum of the (0001) plane GaN.

**Figure 5 materials-16-02255-f005:**
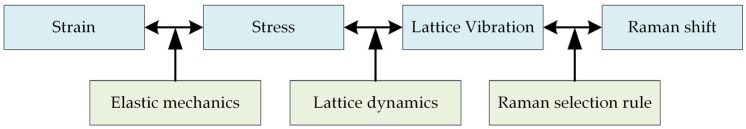
Theoretical framework of the Raman shift–stress relationship model.

**Figure 6 materials-16-02255-f006:**
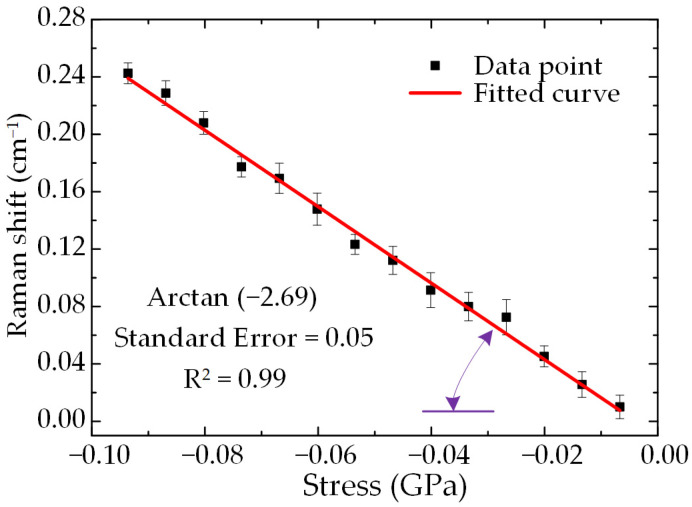
Raman shift of the E_2_^high^ mode for different stress in the calibration experiment.

**Figure 7 materials-16-02255-f007:**
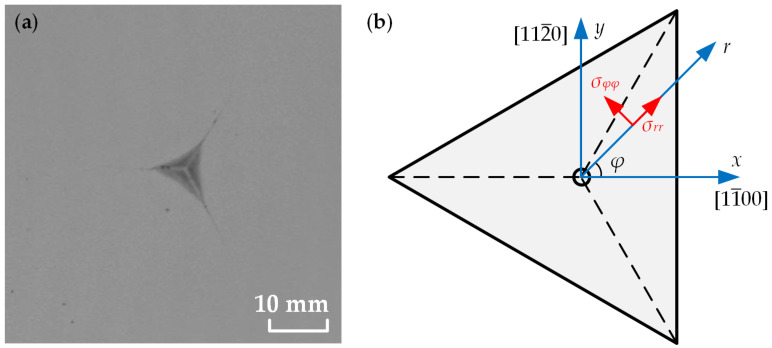
(**a**) Microphotograph of a nanoindentation on the (0001) plane; (**b**) sample, crystal and cylindrical coordinate systems on the (0001) plane.

**Figure 8 materials-16-02255-f008:**
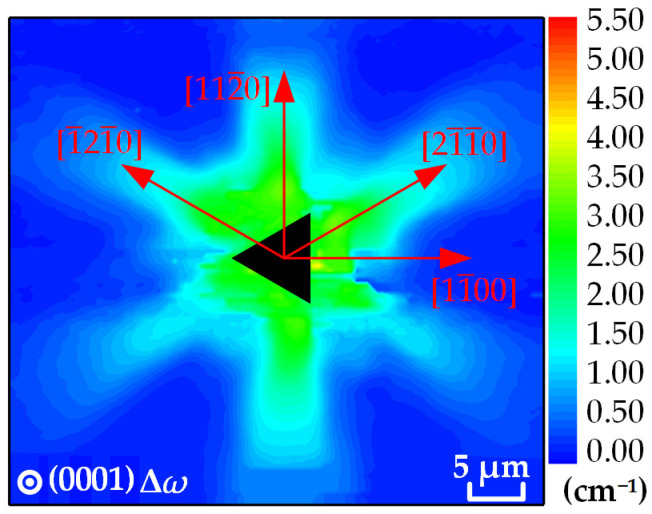
XY mapping of the Raman shift of the E_2_^high^ mode around the indentation of the (0001) plane.

**Figure 9 materials-16-02255-f009:**
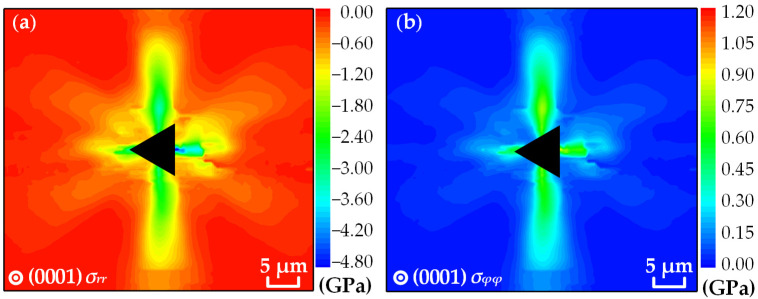
Distributions of (**a**) radial stress *σ_rr_* and (**b**) tangential stress *σ_φφ_*.

**Figure 10 materials-16-02255-f010:**
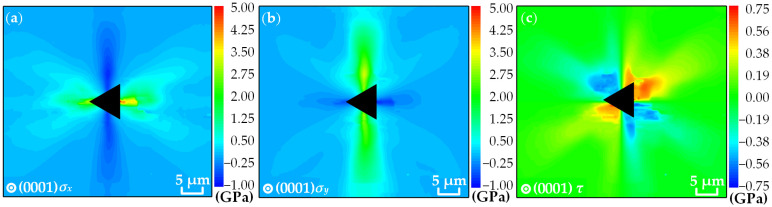
Distributions of (**a**) normal stress *σ_x_*, (**b**) normal stress *σ_y_* and (**c**) shear stress *τ*.

## Data Availability

The data that support the findings of this study are available from the corresponding author upon reasonable request.

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
