# Peer review of "Raman Characterization of the In-Plane Stress Tensor of Gallium Nitride"

_materials, 2023, doi:10.3390/ma16062255_

Round 1

Reviewer 1 Report

The paper used the different Raman modes of single crystalline GaN to separate the stress along different crystallographic direction. It extends the paper of Amilusk [35] by including shear stress, which is not possible for experimentally obtained (0001) films but can occur in other GaN orientations.

However, just saying that the E modes are not sensitive to strain in [0001] direction is wishful thinking. Microscopycally this is immediately clear, because elongating the bonds in [0001] directions will change the lateral bond angles and thus affect the E-modes. In the E-mode vibrations the atoms "rotate" the bonds and that rotation probes also the strain along z direction to some degree. In principle on could extract lateral strain by comparing A mode with E modes.

The same happens with a crystal: Applying compressive stress along sigma_x will result in an expansion of the crystal (and thus strain) along sigma_z (and sigma_y) following Hooke's law. (Unless you use a diamond cell preventing this vertical expansion.)

The result of Fig. 6 should be compared to literature, the paper of Amilusk gives some references and choose factors as well.

Furthermore, eq. 13 should be given directly in terms of the elastic tensor since for (0001) in literature there is usually on one Poission ratio (since strain along [11-20] always has compnenets along [10-10]) and thus conenctions in-plane with [0001] directed strain. Also in the absent of shear stress (like for (0001) oreintation) the cylindrical and hexagonal strain tensor should be similar.

Furthermore, the use of cylindircal coordinate is not so helpful since the strain field are not cylindrical but affected by the crystal axis, could differ in m and a xis for instance on m-plane GaN.

Fig. 8 caption give no description of what mode has been measured respectively what center frequency has been assumed.

Finally, the resultinf fig. 9 does no make sense. How could there be the same radial and tangiental stress? And how comes a two-fold symmetry on a tri-fold symmetric crystal with a trifold symmetric identer derived from a six-fold symmetric Raman shit image? There must be something wrong, that image would work for Si but is plainly wrong for GaN.

And something different: The choice of references. There are lot of reference not related to the paper. Remove 1-4, 7-10 (and use 5 for this) and remove 21-30. In detail:

- Ref. 1-4 are not representative papers that highlite the superiou properties of GaN

- Ref. 5 does not even contain word micro-Raman in the paper. But it indeed compares several methods to measure stress.

- Ref. 6 is ok.

- Ref. 7 is not focussed on stress, and rather covers things like SERS and CARS. Remove. Maybe use ref. 5 instead.

- Ref. 8 seems to be a book chapter. Not really useful as it is not easy to get. Could stay though.

- Ref. 9 is wrong again, it only tells about X-ray diffraction. (And is not high resolution anyway due to the size of typical X-ray bench top foci.)

- Ref. 10 is stimulated Raman and while interesting is of no interest to the current paper.

Ref. 11-19 are silicon and porous silicon which one could do with less citations but ok ...

- Ref. 21-30 are on graphene for which I see no relevance to the current paper.

Ref. 31 is the first using Raman to GaN. Finally!

"Amilusik et al. [35] [...] discovered that the stress generated by the transverse growth of the crystal was much larger than the stress generated by the lattice mismatch." This is wrong. There are different lattice constant in the wing and in the overgrowth region due to doping and defects. A wing has by definition no lattice mismatch as it is free standing ... 

Better "Amilusik et al. [35] characterized the residual stress field of ammonthermal grown GaN on substrates in lateral and vertical direction."

Reviewer 2 Report

Gallium nitride is one of the most popular and promising materials in modern electronics. The development of technologies based on this semiconductor is of strategic importance for industries such as telecommunications, automotive, industrial automation and energy.

Thanks to the implementation of the p-n junction and doping of the transition layer with indium, it was possible to create inexpensive and highly efficient blue and UV LEDs that emit efficiently at room temperature (which is also necessary for laser radiation), this led to the commercialization of high-performance blue LEDs and the long-term life of violet-laser diodes, as well as the development of nitride-based devices such as UV detectors and high-speed field-effect transistors.

Despite the large number of works on this subject, the continuation of research in this direction is extremely relevant.

In this work, a theoretical stress response model for GaN was developed using polarized micro-Raman spectroscopy based on elasticity theory and lattice dynamics. The experimental results showed that the stress component distributions, which differed significantly from the Raman shift distribution, were closely related to the GaN crystal structure and had a gradient along each direction of the crystal.

This study will undoubtedly be useful in the field of design and production of materials based on gallium nitride.

The article is neatly written. All formulas and figures are displayed correctly. References are representative, without inappropriate self-citations.

I definitely recommend this manuscript for publication.

Reviewer 3 Report

line 89: explain the abbreviation " N-GaN"

Reviewer 4 Report

This study deals with the relationship between polarized Raman shift and the different components of stress in a GaN singlecristal that has undergone local modification by nano-indentation. The results presented by these authors are very interesting, but the manuscript can be improved by taking into account these few corrections/additions:

- It would be wise to give the value of the doping of the sample used, as doping also modifies the stress of the material and the latter also varies according to the crystal orientations. Can you provide more information on this subject and detail its impact on your theory?

-It is also mentioned that the dislocation rate is less than 2x10e6 cm-2. As a result, can you discuss the impact of the presence of these dislocations on the stress distribution discussed in this work.

- The laser power used for this study is 20mW. However, this power is very high and the thermal effect on the sample is not negligible and also contributes to the shift observed in the measurements presented. Have you checked if the results remained invariant for powers less than 1mW?

- In Figure 6, the resolution presented is 0.02 cm-1. How is this possible with the equipment used?

Can you give the details of the parameters used with respect to the analytical curve of Figure 6?

Finally, this approach is original in showing the accurate distribution of stress in the (0001) oriented GaN. That being said, as mentioned in the conclusion, this method is not practical for easy decoupling between the observed Raman mode shift and the different components of stress in GaN. Do you have any ideas about how this work could serve Raman users?

Round 2

Reviewer 1 Report

The author addressed most of my concerns. The paper is still hard to understand in some places due to the English and needs some minor corrections.

For instance Line 350:

"If the stress state of the sample was not considered, it would not only give the incorrect trend of stress distribution but also achieve an error of stress value up to 174%" Please consider other wording. It is not clear to me what "stress state" is not to be considered. Do you mean the initial stress state?

Several sentences like this are found throughout the paper.

Same for the figure captions, most have too generic descriptions. For instance, Fig. 6 caption does not describe the figure. "Raman shift of the E2high mode for different stress" would be correct.

And Fig. 8 caption STILL does not tell what mode was measured. Having the reader guess that this is the same mode as in the text pages away for fig. 6 is not helpful. Please write clearly: "XY mapping of the Raman shift of the E2high mode around the indentation of the (0001) plane." and indicate the crystal orientation as in the figure in this cover letter.

The remaining main problem is fig. 9:

Comparing fig 8. to fig. 9 still does not make sense: The Raman shift for the different [11-22] directions halfway out is alternating 1.0 and 2,5 cm-1 with the largest shift at 0°, 60°, 180°, 300° and two weak shifts along 120° and 240°. Thus there is stress in the sample, with either radial or tangential components along these direction. Otherwise there would be no Raman shift. No stress => no Raman shift obviously. HOWEVER, in Figure 9 the radial shift is only observed along 0° and 180° while the shift along 60°, 120°, 240° and 300° is at last 4x times lower and there is no notable tangential stress either. That makes no sense at all. Instead, the radial strain shows strong components along 270°C where the Raman shift was below 0.5 cm-1. Where has the stress gone that induced the large Raman shift along 300° and 60°? Having that shift neither manifest in radial nor tangential stress looks like a fundamental problem. Because there must be stress in these places. At best Fig. 9 is confusing.

In fig. 10 there are negative and positive stresses which can of course somewhat explain this. For publication, I strongly suggest skipping fig. 9 and rather focussing on figure 10.

Did you try the ultimate check and calculated the Raman shift of E2high from figure 10 to give fig. 8 again? (I hope so ... )

Finally, you must state that you ignored any expansion in sigma_z to obtain Fig. 10 in the conclusion (and introduction). But of course, the surface can (and thus will) expands as a reaction to stress in-plane. That would actually strongly reduce the negative contribution along sigma_x in 180°.

Therefore, I strongly suggest that further work include other Raman modes like the A1 modes to get the full stress state and use a rectangular coordinate system. I would like to see also the use of an a-plane or m-plane GaN template/substrate. I could send you a sample of an anisotropic relaxed AlGaN layer on m-GaN which had been thoroughly strain characterised by XRD if you are interested to test your method on anisotropic strain which has all three components and may have local variation close to cracks.
